[Reviews · NeurIPS 2017]

Reviewer 1



Abstract: second sentence doesn’t follow from first one - did authors mean to learn “discrete” instead of “useful” representations? Aside from that first sentence, this is one of the most intriguing and exciting NIPS submissions I’ve ever reviewed. My only question is why this simple-looking approach works so well, when the authors allude to the difficulty others have had in training discrete VAEs. I don’t have direct experience training this sort of model. Is the non-linearity associated with nearest-neighbor lookup so important, as opposed to e.g. greatest vector inner product that would be more typical of neural networks? Also, some discussion of the training time for each data set would have been interesting.

Reviewer 2



Summary- This paper describes a VAE model in which the latent space consists of discrete-valued (multinomial) random variables. VAEs with multinomial latent variables have been proposed before. This model differs in the way it computes which state to pick and how it sends gradients back. Previous work has used sampled softmaxes, and other softmax-based ideas (for example, trying to start with something more continuous and anneal to hard). In this work, a VQ approach is used, where a hard decision is made to pick the closest embedding space vector. The gradients are sent into both into the embedding space and the encoder. Strengths - The model is tested on many different datasets including images, audio, and video datasets. - The model gets good results which are comparable to those obtained with continuous latent variables. - The paper is well-written and easy to follow. Weaknesses- - Experimental comparisons with other discrete latent variable VAE approaches are not reported. It is mentioned that the proposed model is the first to achieve comparable performance with continuous latent variable models, but it would be useful to provide a quantitative assessment as well. - The motivation for having discrete latent variables is not entirely convincing. Even if the input space is discrete, it is not obvious why the latent space should be. For example, the paper argues that "Language is inherently discrete, similarly speech is typically represented as a sequence of symbols" and that this makes discrete representations a natural fit for such domains. However, the latent semantic representation of words or speech could still be continuous, which would make it possible to capture subtle inflections in meaning or tone which might otherwise be lost in a discretized or quantized representation. The applications to compression are definitely more convincing. Overall- The use of VQ to work with discrete latent variables in VAEs is a novel contribution. The model is shown to work well on a multiple domains and datasets. While the qualitative results are convincing, the paper can be made more complete by including some more comparisons, especially with other ways of producing discrete latent representations.

Reviewer 3



The paper introduces a way of doing discrete representation via some discrete transformation in the latent space of a commonly used VAE, and use straight-through estimator to backprop the gradients for learning. The main issue with the paper is its clarity; many notations are used without explanation or used abusively, hence I cannot understand their main equations: In equation (1), could you explain what do you mean by q(z|x) is defined as one hot? Is z a scalar or a one-hot vector? How is it equal to k? In equation (3), what is sg (you never defined it)? Because of the above confusions, I cannot understand what author is proposing. Another problem is the paper has no comparison with any other method in the experimental section. Therefore it’s not really clear whether the technique has any improvement. I will potentially change my score if the author can clearly explain what they’re doing in the rebuttal.